# REASONING-PRESERVED SAFETY ALIGNMENT FOR LARGE REASONING MODELS

## ABSTRACT

Recent research revealed that the reasoning performance of large reasoning models (LRMs) is significantly degraded after safety alignment (i.e., fine-tuning on safety datasets). This phenomenon implies that safety alignment for LRMs has impaired the well-learned parameters crucial for reasoning capabilities. Thus, an interesting question arises: *can we protect the parameters crucial for reasoning from being interfered by safety alignment, thereby acquiring safety capabilities while maintaining the original reasoning capabilities of LRMs?* Motivated by the recent finding that safety capabilities are associated with only a subset of the full parameter space, we propose a novel method that achieves *reasoning-preserved safety alignment* for LRMs. It first identifies reasoning-critical parameters based on a Fisher Information Matrix where each diagonal element represents the importance of the parameter to reasoning capabilities, and then freezes these parameters during fine-tuning on safety datasets. Experiments on multiple reasoning and safety benchmarks validate that our proposed method achieves strong safety performance while maintaining the original reasoning performance of LRMs. Our code is publicly available at `https://anonymous.4open.science/r/RPSA`

## 1 INTRODUCTION

Recent advancements in large reasoning models (LRMs) have showcased their potential in solving complex reasoning tasks, achieving human-level performance on challenging benchmarks (Guo et al., 2025). These developments highlight the growing potential of LRMs as a foundation for tackling challenging tasks (Brown et al., 2020; Touvron et al., 2023). However, deploying LRMs in real-world applications raises serious safety concerns. As LRMs become more powerful, the risks of misuse and unintended consequences increase, making their safety alignment crucial (Jiang et al., 2025; Li et al., 2025a; Huang et al., 2025; Zhou et al., 2025). Current safety alignment for LRMs typically uses supervised fine-tuning or reinforcement learning from human feedback (Bai et al., 2022a; Dai et al., 2024; Ouyang et al., 2022; Wang et al., 2023). However, recent research reveals that safety alignment applied to LRMs often causes substantial degradation in reasoning performance, and this phenomenon is termed *safety tax* (Huang et al., 2025; Xue & Mirzasoleiman, 2025), exposing special challenges for LRMs to acquire safety capacities.

Although some mitigation strategies have been proposed to address the safety tax issue for improving safety capacities of LRMs, they, unfortunately, still degrade reasoning capabilities to a large extent (Huang et al., 2025). For example, the data-centric method that incorporates chain-of-thought reasoning (Jiang et al., 2025) can maintain reasoning performance to some degree, yet often compromises safety capabilities. Besides, the parameter-efficient fine-tuning method of low-rank adaptation (LoRA) (Xue & Mirzasoleiman, 2025) restricts the scope of parameter updates, but this method operates at a coarse granularity without considering the varying importance of individual parameters for reasoning tasks. In summary, these methods suffer from a trade-off between reasoning capabilities and safety capacities in safety alignment for LRMs.

This trade-off actually implies that safety alignment for LRMs has impaired the well-learned parameters crucial for reasoning capabilities. Thus, an interesting question naturally arises:

*Can we protect the parameters crucial for reasoning from being interfered by safety alignment, thereby acquiring safety capabilities while maintaining the original reasoning capabilities of LRMs?*

Motivated by the recent finding that safety capabilities are associated with only a subset of the full parameter space (Wei et al., 2024; Li et al., 2025b; Zhao et al., 2025a), we conjecture that not all parameters need to be updated for achieving safety capacities in safety alignment for LRMs. In this way, when reasoning-critical parameters can be identified and frozen, only updating the remaining parameters in safety alignment is still expected to improve the safety performance for LRMs.

Based on the above insight, we propose a novel method that selectively freezes reasoning-critical parameters to achieve *reasoning-preserved safety alignment* for LRMs. Concretely, our method first estimates the importance of each parameter for reasoning capabilities according to the diagonal elements in the Fisher Information Matrix derived from a reasoning corpus. It then ranks parameters by their Fisher values and freezes the top-$k$ fraction during safety fine-tuning, where $k$ is dynamically adjusted during the training process based on the gradient conflict between reasoning and safety objectives. This adaptive freezing strategy ensures that parameters most critical for reasoning remain protected when conflicts arise, while allowing more flexibility when gradients align. We apply this dynamic freezing approach to both full fine-tuning and LoRA-based parameter-efficient training, with parameters selectively frozen according to their Fisher importance scores. In comprehensive evaluations, our method substantially reduces the safety tax, maintaining reasoning performance close to the original model while achieving safety comparable to standard alignment methods.

## 2 RELATED WORK

**LLM/LRM Alignment and Safety Training.** Mainstream post-training alignment methods optimize output preferences through various approaches: supervised instruction tuning (Guo et al., 2025; Jaech et al., 2024; Bianchi et al., 2024), InstructGPT's Proximal Policy Optimization with reward models (Ouyang et al., 2022), Constitutional AI's AI-generated harmlessness feedback (Bai et al., 2022b), Self-Instruct's synthetic instructions (Wang et al., 2023), Direct Preference Optimization's direct preference objectives (Rafailov et al., 2023), and Safe-RLHF's safety-constrained reward maximization (Dai et al., 2024). Recent work explores safety-aware fine-tuning (Choi et al., 2024), multi-round red-teaming (Ge et al., 2024), and prospect-theoretic alignment objectives (Ethayarajh et al., 2024). Guan et al. (2024) proposes deliberative alignment linking safety and reasoning through integrated processes. However, how safety gradients interact with reasoning critical weights during training remains underexplored. These methods primarily operate on output distributions or objective functions without considering parameter-level importance for reasoning, motivating our parameter-aware protection approach.

**Safety Tax and Reasoning–Safety Trade-off.** Recent research (Huang et al., 2025; Zhou et al., 2025) revealed the *safety tax* phenomenon where safety alignment significantly degrades reasoning capabilities in LRMs. Safety guardrails prove brittle under pruning, low-rank modifications, and downstream fine-tuning (Xue & Mirzasoleiman, 2025; Huang et al., 2025). Current mitigations operate at various levels: data-centric approaches like SafeChain incorporate chain-of-thought into safety datasets (Jiang et al., 2025), though this can increase attack vulnerability; mechanistic methods localize safety-critical layers for selective freezing (Li et al., 2025b); and parameter-efficient approaches employ LoRA for constrained updates (Xue & Mirzasoleiman, 2025). However, these methods do not directly identify and protect parameters based on reasoning importance.

## 3 METHODOLOGY

In this section, we propose **R**easoning-**P**reserved **S**afety **A**lignment (RPSA), a method for LRMs that selectively freezes reasoning-critical parameters to achieve safety alignment without sacrificing reasoning ability. We resolve this by computing importance scores using the diagonal Fisher Information Matrix (FIM) on a reasoning corpus to identify parameters most essential for reasoning. Based on these scores, our method constructs a protection mask that selectively freezes reasoning-critical parameters during safety fine-tuning, with the frozen ratio dynamically adjusted based on gradient conflicts between reasoning and safety objectives. This adaptive freezing strategy ensures that parameters most critical for reasoning remain protected when conflicts arise, while allowing

more flexibility when gradients align. This targeted method preserves the model's reasoning circuitry while enabling effective safety alignment, significantly reducing the reasoning degradation that typically occurs after safety training.

## 3.1 QUANTIFYING PARAMETER IMPORTANCE WITH EMPIRICAL FISHER INFORMATION

Assume that we have a LRM $f_{\boldsymbol{\theta}_R}$ with parameters $\boldsymbol{\theta}_R$, which has been optimized on training dataset $\mathcal{D}_R$ for reasoning tasks. Our goal is to obtain a new set of parameters $\boldsymbol{\theta}_S$ by fine-tuning on safety dataset $\mathcal{D}_S$, such that the safety loss $\mathcal{L}_S$ is minimized while the reasoning loss $\mathcal{L}_R$ is preserved. To understand how changes in parameters affect reasoning performance, we consider a second-order Taylor expansion of the reasoning loss $\mathcal{L}_R(\boldsymbol{\theta})$ around the optimal reasoning parameters $\boldsymbol{\theta}_R^*$:

$$\mathcal{L}_R(\boldsymbol{\theta}) \approx \mathcal{L}_R(\boldsymbol{\theta}_R^*) + (\boldsymbol{\theta} - \boldsymbol{\theta}_R^*)^\top \nabla_{\boldsymbol{\theta}} \mathcal{L}_R(\boldsymbol{\theta}_R^*) + \frac{1}{2}(\boldsymbol{\theta} - \boldsymbol{\theta}_R^*)^\top \boldsymbol{H}(\boldsymbol{\theta} - \boldsymbol{\theta}_R^*), \tag{1}$$

where $\boldsymbol{\theta} \in \mathbb{R}^d$ is the vector of model parameters, $\nabla_{\boldsymbol{\theta}} \mathcal{L}_R(\boldsymbol{\theta}_R^*) \in \mathbb{R}^d$ is the gradient of the reasoning loss evaluated at $\boldsymbol{\theta}_R^*$, and $\boldsymbol{H} = \nabla_{\boldsymbol{\theta}}^2 \mathcal{L}_R(\boldsymbol{\theta}_R^*) \in \mathbb{R}^{d \times d}$ is the Hessian matrix. Since $\boldsymbol{\theta}_R$ is around the optimal $\boldsymbol{\theta}_R^*$, the gradient term can be neglected, and the increase in loss is dominated by the quadratic term. Computing the exact Hessian $\boldsymbol{H}$ is infeasible for large models. A common alternative is to approximate it using the Fisher Information Matrix (FIM) (Kunstner et al., 2019). For autoregressive models, the FIM can be written token-wise as:

$$\boldsymbol{F} = \mathbb{E}_{(x,y_{<t}) \sim p(\boldsymbol{x}, \boldsymbol{y}_{<t})} \left[ \mathbb{E}_{y_t \sim p_{\boldsymbol{\theta}}(y_t | \boldsymbol{x}, \boldsymbol{y}_{<t})} \left[ \nabla_{\boldsymbol{\theta}} \log p_{\boldsymbol{\theta}}(y_t \mid \boldsymbol{x}, \boldsymbol{y}_{<t}) \nabla_{\boldsymbol{\theta}} \log p_{\boldsymbol{\theta}}(y_t \mid \boldsymbol{x}, \boldsymbol{y}_{<t})^\top \right] \right], \tag{2}$$

where $(x, y_{<t})$ is the input context (original input plus previously generated tokens), and $y_t$ is the current token output. For computational efficiency, we focus on the diagonal elements of $\boldsymbol{F}$:

$$F_{i,i} = \mathbb{E}_{(\boldsymbol{x}, \boldsymbol{y}_{<t}) \sim p(\boldsymbol{x}, \boldsymbol{y}_{<t})} \left[ \mathbb{E}_{y_t \sim p_{\boldsymbol{\theta}}(y_t | \boldsymbol{x}, \boldsymbol{y}_{<t})} \left[ \left( \frac{\partial}{\partial \theta_i} \log p_{\boldsymbol{\theta}}(y_t \mid \boldsymbol{x}, \boldsymbol{y}_{<t}) \right)^2 \right] \right]. \tag{3}$$

Computing the expectation over the entire input space is practically impossible, since the space is effectively infinite. Usually, this expectation is approximated using the model's training data for $\boldsymbol{x}, \boldsymbol{y} < t$ and $y_t$, which is known as the empirical Fisher. However, most LRMs do not publicly release their training data. To work around this, we instead use a Monte Carlo approach: we collect model responses, including chain-of-thought (CoT) reasoning steps, from the MATH-500 dataset (Lightman et al., 2023), to construct a reference dataset $\mathcal{D}_{\text{ref}}$. Then for each sample $(\boldsymbol{x}, \boldsymbol{y})$ in $\mathcal{D}_{\text{ref}}$, we perform backpropagation to compute the squared gradients with respect to each parameter $\theta_i$, providing a Monte Carlo estimate of the Fisher diagonal:

$$\hat{F}_{i,i} = \frac{1}{|\mathcal{D}_{\text{ref}}|} \sum_{(\boldsymbol{x}, \boldsymbol{y}) \in \mathcal{D}_{\text{ref}}} \frac{1}{|\boldsymbol{y}|} \sum_{t=1}^{|\boldsymbol{y}|} \left( \frac{\partial}{\partial \theta_i} \log p_{\boldsymbol{\theta}}(y_t \mid \boldsymbol{x}, \boldsymbol{y}_{<t}) \right)^2, \tag{4}$$

After approximating the Hessian $\boldsymbol{H}$ of the reasoning loss with the diagonal of the Fisher Information Matrix $\boldsymbol{F}$, the second-order term of $\mathcal{L}_R(\boldsymbol{\theta})$ can be expressed as

$$\frac{1}{2}(\boldsymbol{\theta} - \boldsymbol{\theta}_R^*)^\top \boldsymbol{H}(\boldsymbol{\theta} - \boldsymbol{\theta}_R^*) \approx \frac{1}{2} \sum_{i=1}^d \hat{F}_{i,i}(\theta_i - \theta_{R,i}^*)^2. \tag{5}$$

This formulation indicates that the contribution of each parameter $\theta_i$ to the increase in reasoning loss is proportional to both its squared deviation $(\theta_i - \theta_{R,i}^*)^2$ and the corresponding diagonal Fisher entry $\hat{F}_{i,i}$. Parameters associated with larger $\hat{F}_{i,i}$ values are more sensitive, such that even small perturbations can lead to substantial increases in $\mathcal{L}_R$, whereas parameters with smaller $\hat{F}_{i,i}$ can be adjusted with comparatively minor impact on reasoning performance. In this sense, the diagonal Fisher entries quantify the importance of parameters in preserving the model's reasoning capability.

## 3.2 PROTECTING REASONING CAPABILITIES DURING ALIGNMENT

We adopt a parameter freezing strategy to preserve reasoning capabilities during safety fine-tuning by freezing the most important parameters. However, when many parameters $\theta_i$ have $\hat{F}_{i,i} = 0$, it

is impossible to establish their relative importance. Therefore, we restrict the freezing operation to parameters with non-zero $\hat{F}_{i,i}$ only. Let the *frozen ratio* be a hyperparameter $k \in [0, 1]$. We first compute the corresponding percentile threshold: $\tau_k = Q_{1-k}(\{F_i \mid F_i > 0\})$, where $Q_{1-k}(\cdot)$ denotes the $(1 - k)$-th percentile computed over parameters with non-zero Fisher values. A binary mask is then defined as $M_i = \mathbf{1}[F_i \leq \tau_k]$. During fine-tuning, gradients are modified by this mask, $\tilde{g}_i = M_i \cdot g_i$, so that parameters with $M_i = 0$ are frozen ($\tilde{g}_i = 0$), ensuring that reasoning-critical parameters remain unchanged while adaptation for safety occurs only in the remaining parameters.

This Fisher-based freezing method can be applied to different fine-tuning paradigms. In full fine-tuning (FFT), Fisher importance is computed for all trainable parameters $\boldsymbol{\theta}$, and the top-$k$ fraction of parameters among those with non-zero Fisher values are frozen. In parameter-efficient fine-tuning with LoRA, where the base model parameters are already frozen, Fisher importance is computed directly on the initialized LoRA parameters $\boldsymbol{\theta}_{\text{LoRA}}$, and the top-$k$ fraction of LoRA parameters with non-zero Fisher values are selectively frozen.

### 3.3 Dynamic Freezing Strategy

A key challenge of our method lies in the choice of the hyperparameter $k$. The optimal value of $k$ may vary across different models, and exhaustively searching for it can be both time-consuming and computationally expensive. To mitigate this issue, we introduce an adaptive approach that dynamically updates $k$ during training based on gradient information, providing a flexible alternative to manual hyperparameter tuning.

During training, we introduce two datasets, the reasoning dataset and the safety dataset, and compute gradients from each, denoted by $\boldsymbol{g}_R, \boldsymbol{g}_S \in \mathbb{R}^d$. In practice, we use the same $\mathcal{D}\text{ref}$ used in the FIM estimation as the reasoning dataset. The gradient $\boldsymbol{g}_S$ corresponds to the unmasked gradient obtained during the safety fine-tuning process. For each parameter $\theta_j$, the associated gradients are denoted by $g_{R,j}$ and $g_{S,j}$. We then define the conflict measure $c$ as the proportion of parameters with non-zero $\hat{F}_{i,i}$ whose gradients have opposing signs, computed only over parameters with non-zero Fisher values. This is because only parameters with non-zero Fisher values can be frozen; parameters with zero Fisher are unaffected by our method and thus irrelevant for the conflict measure. Formally, the conflict is given by

$$c = \frac{\left|\{j \mid \hat{F}_{j,j} > 0 \ \wedge \ g_{R,j} g_{S,j} < 0\}\right|}{\left|\{j \mid \hat{F}_{j,j} > 0\}\right|}. \tag{6}$$

We keep a sliding window of size $W$ to record the recent history of the conflict values $c$. For the first $W$ steps, when there is insufficient history, we set $k_{\text{dyn}} = 0.5$. Afterward, we compute the mean $\mu$ and standard deviation $\sigma$ of the past $W$ conflict values to serve as a baseline. The current conflict value is normalized relative to this baseline and mapped into $[0, 1]$ to obtain $k_{\text{dyn}}$:

$$k_{\text{dyn}} = \text{clip}\left(\frac{c - \mu}{2\sigma} + 0.5, \ 0, \ 1\right), \tag{7}$$

where $\text{clip}(\cdot, 0, 1)$ truncates the result into the $[0, 1]$ range. Note that we use $2\sigma$ as the scaling factor, since under the normal distribution assumption about 95% of the samples fall within $[\mu - 2\sigma, \mu + 2\sigma]$. This provides a natural interval for mapping $c$ into the range $[0, 1]$. This makes the frozen ratio adaptive: lower conflict decreases $k_{\text{dyn}}$ to update more parameters, while higher conflict increases $k_{\text{dyn}}$ to freeze more and preserve reasoning capacity.

## 4 Experiments

### 4.1 Identifying Reasoning-Critical Parameters with Fisher Information

To approximate the original training distribution, we adopt a reference dataset $\mathcal{D}_{\text{ref}}$. Specifically, we sample 100 problems from the **MATH-500** (Lightman et al., 2023), which provides a diverse set of mathematical reasoning tasks that are representative of the complex, multi-step logical processes LRMs are designed to handle. These problems are fed into the target models, and both the final answers and the associated CoT traces are collected. The resulting corpus serves as a surrogate reasoning dataset, enabling the estimation of Fisher information for identifying parameters that are critical to reasoning performance.

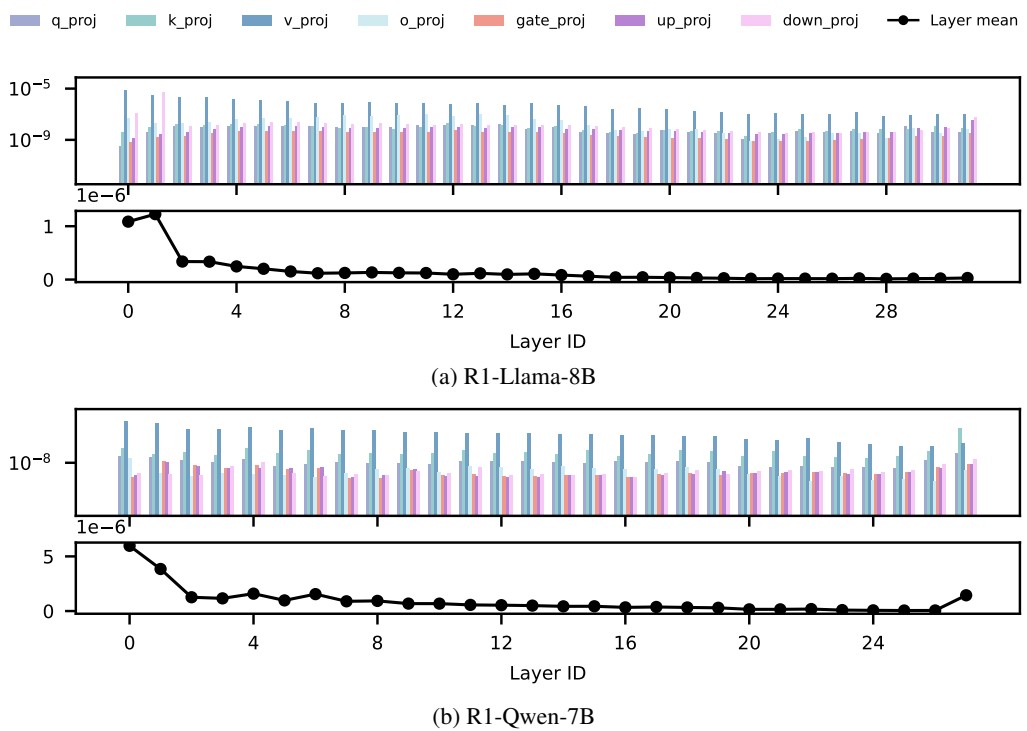

Figure 1: Fisher information distribution across layers and modules. We report the mean Fisher information for parameters grouped by their module and layer index. In each sub-figure, the upper panel illustrates the Fisher information of different modules within each layer, the lower panel summarizes the mean Fisher information at the layer level.

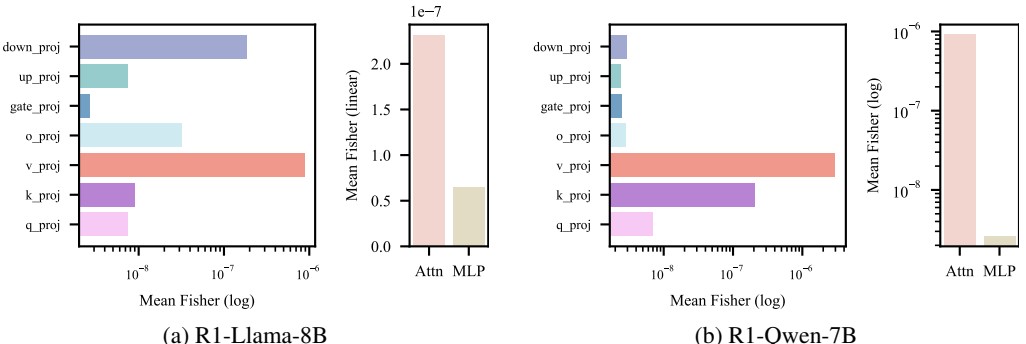

Figure 2: Mean Fisher information across layers for each parameter matrix and the overall mean Fisher information for each module. In each subfigure, the left panel displays the mean Fisher information of individual parameter matrices, while the right panel compares the overall mean Fisher information for the Attention and MLP modules.

Using this corpus to estimate diagonal Fisher information for the reasoning loss, we find a highly non-uniform layer-wise distribution as shown in Figure 1. We highlight two important observations:

**Reasoning-critical parameters are concentrated in the early and final layers of the model.** Previous study Zhao et al. (2025b) argues that the reasoning capability of large language models mainly emerges in their last few layers, often associating complex reasoning with progressively abstract and context-sensitive representations. However, our Fisher-based analysis provides a distinct perspective. As shown in Figure 1, the layers with the highest average Fisher information are the very first layers, followed by the final layers of the network. While the last layers exhibit high

Fisher scores in Qwen (1b), this pattern is not observed in LLaMA (1a); moreover, in Qwen, the mean Fisher of the last layers is lower than that of the initial layers. These observations suggest that early layers may play a crucial role in preparing, refining, or filtering representations that are essential for subsequent reasoning. This insight could inform more efficient model training and fine-tuning strategies by emphasizing the critical role of early layers.

**Attention modules are critical for reasoning.** In Figure 2, our Fisher-based analysis shows large module-wise differences. Across all modules, the Fisher value of the *value* projection matrix $V$ is the largest, and this trend is consistently observed in both models. From the perspective of the mean values across the two models, the *key* projection matrix $K$ also holds a significant position. This may indicate that the abilities of $K$ and $V$ to model token-to-token relationships and to aggregate information are crucial for reasoning. Although certain feed-forward components such as the *up* and *down* projections also show elevated Fisher values, their importance appears secondary compared with the dominant contribution of attention modules. This observation is further echoed by our experiments in Section 4.3, where applying LoRA to attention modules led to larger performance drops. Overall, this contrast suggests that reasoning capability in LLMs is disproportionately grounded in attention rather than evenly distributed across modules.

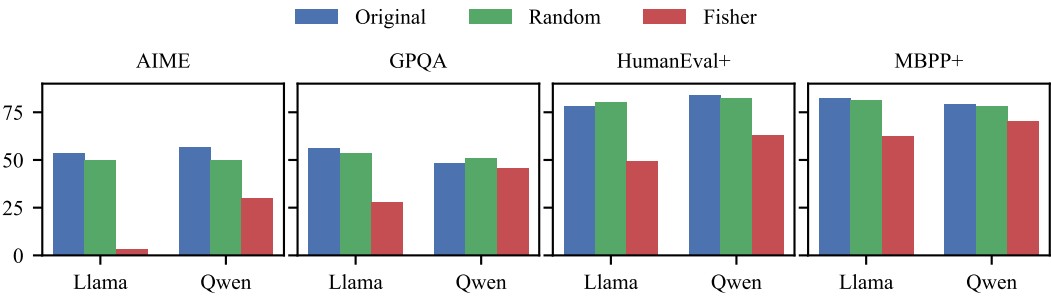

Figure 3: Evaluation results on AIME and GPQA datasets after adding perturbations to the model. Fisher-guided perturbations consistently underperform random perturbations.

**Verifying the Effectiveness of Fisher Information in Identifying Reasoning-Critical Parameters.** We designed a straightforward experiment based on the hypothesis that if parameters with high Fisher information are indeed critical for reasoning, then perturbing these parameters should lead to a larger decline in the model's reasoning performance compared to perturbing other parameters. To test this, we ranked all model parameters according to their Fisher information values and applied a $0.9\times$ scaling to the top 1% of parameters, and for comparison, we also applied the same scaling to 1% of randomly selected parameters. In both cases, only a small fraction of parameters were perturbed, ensuring that the overall network structure remained largely intact. The impact of these perturbations on the model's reasoning ability was evaluated using problems from the 2024 AIME (American Invitational Mathematics Examination) and the Main subset of the GPQA (Rein et al., 2024) benchmark, representing high-level reasoning tasks, as well as the HumanEval+ and MBPP+ (Liu et al., 2023) datasets, which focus on programming tasks that test the model's coding and problem-solving abilities. As shown in Figure 3, perturbing the Fisher-top 1% caused a markedly larger drop than random perturbation, supporting that high-Fisher parameters are critical for reasoning and validating Fisher as a practical importance estimator.

## 4.2 MITIGATING THE SAFETY TAX WITH SELECTIVE PARAMETER FREEZING

In our main experiment, we apply our proposed RPSA method to two commonly used safety fine-tuning paradigms: full fine-tuning (FFT) and LoRA. We first use 100 samples from the MATH-500 dataset to compute the FIM for both the full model parameters and the LoRA adapters attached to the model. We set the window size of dynamic $k$ strategy to $W = 20$. In implementation, we compute the reasoning gradient $g_R$, the conflict ratio $c$, and the corresponding $k_{\text{dyn}}$ every 25 training steps, and recompute the gradient mask $M$ accordingly. Under the LoRA setting, we use $r = 4$, $\alpha = 16$,

and a dropout rate of 0.05. LoRA is applied exclusively to the MLP modules. In Section 4.3, we discuss the effect of different ranks $r$ on the performance of RPSA, and the impact of applying LoRA to different modules. Detailed experimental settings can be found in Appendix A.

### 4.2.1 EXPERIMENTAL SETUP

**Models.** We conduct experiments on distilled lightweight LRMs from the DeepSeek-R1 series (Guo et al., 2025), specifically *DeepSeek-R1-Distill-Llama-8B* (R1-Llama-8B) and *DeepSeek-R1-Distill-Qwen-7B* (R1-Qwen-7B). These distilled models preserve the core reasoning capabilities and overall performance of their larger counterparts while offering a more compact architecture.

**Datasets.** To evaluate reasoning capabilities, we conduct experiments on the 2024 AIME, the Main subset of GPQA (Rein et al., 2024), HumanEval+ (HE+), and MBPP+ (Liu et al., 2023) benchmarks. AIME and GPQA focus on challenging reasoning tasks, with AIME targeting mathematical problem solving and structured derivations, and GPQA emphasizing multi-step logical thinking and common-sense reasoning. HumanEval+ and MBPP+ evaluate code generation and programming reasoning, measuring the model's ability to produce correct and functional programs. For safety evaluation, we use the StrongReject (Souly et al., 2024) and WildJailbreak (Jiang et al., 2024) datasets. For reasoning benchmarks, we report Pass@1 accuracy to measure correctness. For safety benchmarks, we report the Attack Success Rate (ASR), which quantifies the harmfulness of the model.

**Evaluation.** All evaluations follow the recommended configurations in Guo et al. (2025) and are conducted using the *lm-evaluation-harness* framework (Gao et al., 2024). To accommodate CoT reasoning, we modify the prompt templates and answer extraction logic. The model is evaluated with a temperature of 0.6 and top-p sampling of 0.95 to reduce repetitive or incoherent outputs. All instructions are provided directly in the user prompt without any system prompt. For CoT reasoning, prompts explicitly include the phrase `Please think step by step` and the `<think>` prefix. For mathematical problems, prompts further instruct the model to reason step by step and enclose the final answer in the form of `\boxed{answer}`. Prompt templates are also adjusted to comply with official guidelines, with detailed templates provided in Appendix B.

### 4.2.2 MAIN RESULTS

The results of our experiments, visualized in Figure 4 and detailed in Table 1, demonstrate that our proposed method effectively mitigates the safety tax. Figure 4 shows the trade-off between reasoning performance (x-axis) and safety, measured by the average StrongReject and WildJailbreak ASR (y-axis, inverted so higher positions indicate lower ASR), with the upper-right corner representing models with both high reasoning performance and high safety. To quantify this trade-off more concretely, we additionally define a *Safety Tax Rate (STR)*, measured as $STR = \Delta R / \Delta S$, where $\Delta R$ denotes the average performance drop across all reasoning benchmarks, and $\Delta S$ denotes the average reduction in ASR across all safety benchmarks. Intuitively, STR captures how much reasoning ability is lost per unit of safety gain: lower values indicate a more favorable trade-off.

Across all four subplots, the Original models (blue circles) establish a baseline of strong reasoning ability but poor safety, reflected in high ASR scores. Standard alignment methods expose the safety tax: FFT (green triangles) nearly eliminates ASR but causes a catastrophic drop in reasoning, while LoRA (orange triangles) achieves a milder trade-off with improved reasoning ability. In contrast, our proposed RPSA method further mitigates the reduction on the reasoning ability, with lower average harmfulness in most cases.

As shown in Table 1, our method significantly reduces STR under different settings on the R1-Llama-8B model. Applied to the baseline FFT, it lowers STR from 0.9221 to 0.5837, achieving a maximum reduction of 36.70%. When applied to LoRA, it further decreases STR from 0.2354 to 0.0834, corresponding to a 64.57% reduction. Overall, combining our method with LoRA yields a reduction from 0.9221 (original FFT) to 0.0834, representing a 90.96% decrease. On the R1-Qwen-7B model, our approach achieves up to a 43.39% reduction in STR, demonstrating its consistent and substantial effectiveness.

It is worth noting that we employ MATH-500, a mathematics-domain dataset, as a reference dataset for computing the FIM. Nevertheless, when evaluating the model fine-tuned with our method on

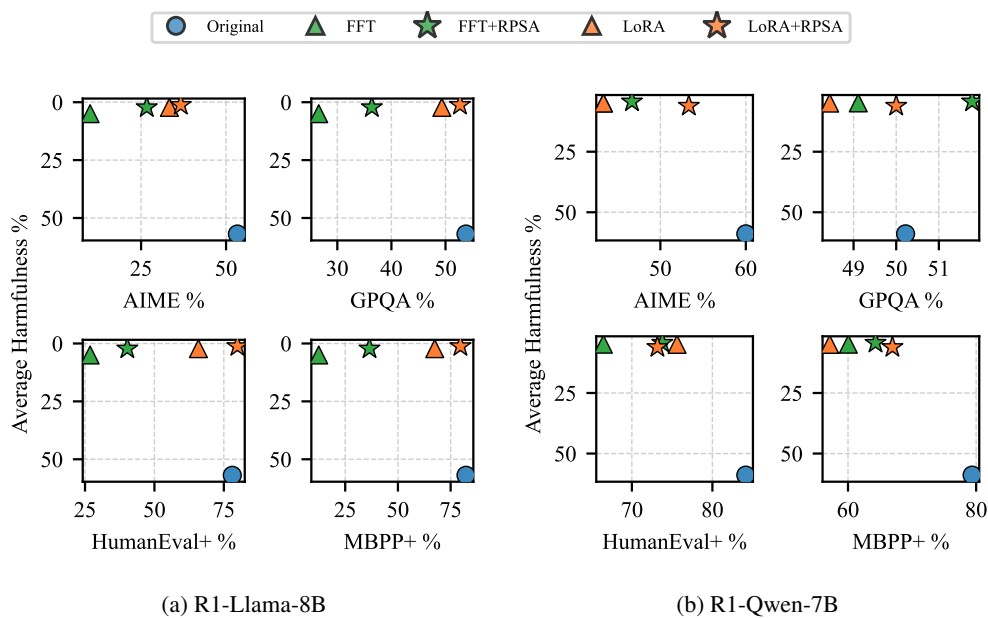

(a) R1-Llama-8B  (b) R1-Qwen-7B

Figure 4: Results for different methods on various datasets. Please note that LoRA and FFT overlap at the same spot in the AIME subfigure of R1-Qwen-7B, with only a small difference along the y-axis. The same color denotes the same base fine-tuning method (FFT/LoRA). Our proposed method, RPSA, is highlighted with star markers.

Table 1: Experimental comparison of our RPSA method with baseline methods. The best performance in each group is highlighted in bold.

| Model | Method | Reasoning Performance ↑ | | | | Harmfulness ↓ | | STR↓ |
|-------|--------|------|------|-----|-------|-----|-----|-----|
| | | AIME | GPQA | HE+ | MBPP+ | SR | WJ | |
| R1-Llama-8B | Original | 53.33 | 53.79 | 78.05 | 82.28 | 64.22 | 49.60 | N/A |
| | FFT | 10.00 | 26.56 | 26.83 | 12.43 | 6.71 | 3.20 | 0.9221 |
| | **+RPSA** | **26.67** | **36.38** | **40.24** | **36.51** | **2.87** | **1.60** | **0.5837** |
| | LoRA | 33.33 | 49.33 | 65.85 | 67.46 | 1.28 | 3.20 | 0.2354 |
| | **+RPSA** | **36.67** | **52.68** | **79.88** | **79.63** | **0.00** | 2.40 | **0.0834** |
| R1-Qwen-7B | Original | 60.00 | 50.22 | 84.15 | 79.37 | 64.22 | 53.60 | N/A |
| | FFT | 43.33 | 49.11 | 66.46 | 60.05 | 1.28 | 8.40 | 0.2533 |
| | **+RPSA** | **46.67** | **51.78** | **73.78** | **64.29** | **0.63** | **8.00** | **0.1704** |
| | LoRA | 43.33 | 48.44 | **75.61** | 57.14 | **0.64** | 9.20 | 0.2279 |
| | **+RPSA** | **53.33** | **50.00** | 73.17 | **66.93** | 0.96 | 11.20 | **0.1434** |

HumanEval+ and MBPP+, two code generation benchmarks, we also observe comparable improvements. This indicates that our identification of reasoning-critical parameters exhibits strong cross-domain generalization, thereby underscoring the practicality of our approach.

### 4.3 HYPERPARAMETER ANALYSIS

We analyze three knobs that affect the safety–reasoning tradeoff: LoRA rank, LoRA target modules, and the frozen ratio $k$. All experiments in this subsection are conducted on DeepSeek-R1-Distill-Llama-8B.

**LoRA Rank.** The rank of LoRA adapters is crucial as it determines the sparsity of parameter updates. According to Xue & Mirzasoleiman (2025), We performed a hyperparameter search over

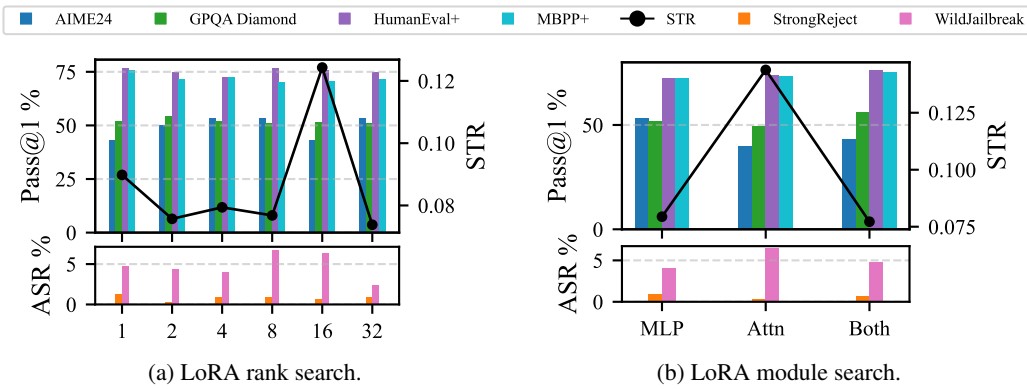

(a) LoRA rank search.

(b) LoRA module search.

Figure 5: Hyperparameter search for LoRA rank and LoRA module on the Llama-8B model under constant frozen ratio $k = 0.6$. Results on plain LoRA can be found in Appendix C

the LoRA rank. Figure 5a presents the results for ranks ranging from 1 to 32 under a constant $k = 0.6$. Most settings yield consistently low STR (e.g., ranks 2, 4, 8, 32). For consistency with prior work (Xue & Mirzasoleiman, 2025), we adopt rank = 4 in our main experiments.

**LoRA Module.** We investigated the effect of applying LoRA to different modules, considering three setups: MLP-only, Attention-only, and Both. We conducted fine-tuning under a constant $k = 0.6$, and the results are shown in Figure 5b. Applying LoRA to only the Attention module leads to a degradation in reasoning performance, which is consistent with our earlier analysis in Section 4.1. In contrast, applying LoRA to the MLP modules, or to both MLP and Attention, preserves almost the same average reasoning performance as the MLP-only setting. Despite the lowest STR observed for MLP & Attention, we adopt MLP-only as the default target for LoRA in our main experiments, both for consistency with prior work and computational efficiency.

## 4.4 ABLATION STUDY

We compared Fisher-guided masking with a random masking baseline under the same constant frozen ratio $k = 0.6$ using the FFT paradigm. In the Fisher method, we identify all parameters with non-zero Fisher values, rank them, and freeze the top 60%. For the random baseline, we freeze the same number of parameters sampled uniformly at random. We then operate the safety fine-tuning on both models. As shown in Figure 6, models frozen with Fisher guidance consistently achieve higher STR than those trained with either no masking or random masking. While the random mask results in a decrease in STR com-

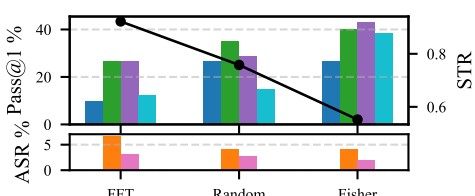

Figure 6: Comparison of Fisher-Guided mask, Random mask and FFT with no masking. The legend is the same as Figure 5

pared to the original FFT, it still underperforms the Fisher-guided method. These results highlight the effectiveness of our proposed Fisher-guided approach.

## 5 CONCLUSION

In summary, we introduced a Fisher-information-based method to safeguard reasoning parameters during safety alignment, effectively mitigating the degradation of reasoning ability in large reasoning models. Our experiments demonstrate that this approach achieves a more favorable balance between safety and reasoning compared with existing methods. To further reduce the computational overhead from hyperparameter search, we incorporate an adaptive strategy to automatically adjust the freezing proportion, ensuring that our framework remains both practical and flexible across diverse settings.

## ETHICS STATEMENT

Our experiments involve publicly available datasets and model outputs that may contain harmful or biased content prior to safety fine-tuning. Such content is used only for analysis in controlled settings, and we do not release or promote it beyond the scope of this research.

## REPRODUCIBILITY STATEMENT

We provide detailed descriptions of our methods, training procedures, and evaluation settings in the main paper, with hyperparameters and training settings documented in Appendix A. All datasets used are publicly available, and their sources are explicitly specified. The URL for our code is provided in the abstract, and the code files are included in the supplementary materials.

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

# A  TRAINING DETAILS

Table 2: We divide our experiments into two categories: Full fine-tuning and LoRA-based fine-tuning. For a fair comparison, both settings are conducted under the same base hyperparameter configuration, unless otherwise specified. All reported results are obtained under this unified setup.

| Hyperparameter | Full fine-tuning | LoRA |
|---|---|---|
| Learning rate | $1 \times 10^{-5}$ | $1 \times 10^{-4}$ |
| Epochs | 3 | 6 |
| Per Device Batch size | 2 | 2 |
| Precision | bfloat16 | bfloat16 |
| Optimizer | AdamW | AdamW |
| Weight decay | $1 \times 10^{-4}$ | $1 \times 10^{-4}$ |
| Gradient clipping | 1.0 | 1.0 |
| Scheduler | Cosine, warmup 100 steps | Cosine, warmup 100 steps |
| Eval steps | 100 | 100 |
| Save steps | 100 (keep best) | 100 (keep best) |
| Random seed | 42 | 42 |
| Hardware | $4 \times$ H100 GPUs | $2 \times$ H100 GPUs |
| LoRA config | — | $r = 4, \alpha = 16, \text{dropout} = 0.05$ |
| Frameworks | PyTorch + Hugging Face Transformers + vLLM | |

# B  EVALUATION PROMPTS

---

**AIME Prompt**

Solve the following math problem. Present the final answer in the format: Final Answer: boxed{your_answer}.
Problem: {{problem}}
Answer:

---

**GPQA Prompt**

What is the correct answer to this question:{{Question}}
Choices:
(A) {{choice1}}
(B) {{choice2}}
(C) {{choice3}}
(D) {{choice4}}
Please reason step by step, and present your final answer using the format: Final Answer: boxed{X}, where X is one of A, B, C, or D.

---

**HumanEval Prompt**

Solve the following problem.

Problem:
{{ prompt }}

Please reason step by step, and put your final answer within a code block:

---

---

**MBPP+ Prompt**

{{prompt if prompt is defined else text}} Your code should satisfy the following assertion:
{{test_list[0]}}. Please reason step by step, and put your final answer within a code block:

---

## C   ADDITIONAL EXPERIMENTAL RESULTS

In this section, we further conducted a comprehensive hyperparameter search for the standalone
LoRA module, as illustrated in Figure 7. The experimental results indicate that, except for the case
where $r = 16$ on MLP modules, our method consistently outperforms plain LoRA across most
configurations. This suggests that our approach exhibits a higher level of robustness and effective-
ness under varying hyperparameter settings. The observed performance gains further highlight the
advantages of our method over the standard LoRA implementation.

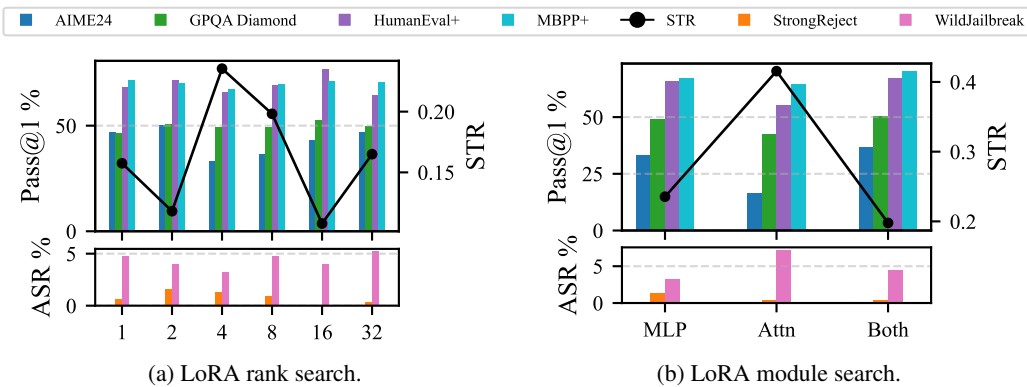

Figure 7: Hyperparameter search for LoRA rank and LoRA module on the Llama-8B model with
plain LoRA.

## D   USE OF LARGE LANGUAGE MODELS

Large language models were employed exclusively to improve the clarity, grammar, and readability
of the manuscript. They were not used for research ideation, experiment design, data analysis, or
retrieval of related work.

