# OpenReview forum: "Reasoning-Preserved Safety Alignment for Large Reasoning Models"
_ICLR.cc/2026/Conference — ICLR 2026 Conference Withdrawn Submission_

### Official Review · Reviewer_bc4t · 2025-10-29

**Soundness:** 3
**Presentation:** 3
**Contribution:** 2
**Rating:** 4
**Confidence:** 5

**Summary:**

This paper proposes Reasoning-Preserved Safety Alignment (RPSA), a fine-tuning method for large reasoning models (LRMs) that aims to mitigate the safety tax—the degradation of reasoning performance during safety alignment. The key idea is to compute diagonal Fisher Information from a reasoning corpus to identify “reasoning-critical” parameters, which are then selectively frozen during safety fine-tuning. The method further introduces a dynamic freezing mechanism that adjusts the frozen ratio based on gradient conflicts between reasoning and safety objectives. Experiments on DeepSeek-R1 models show improved reasoning–safety trade-offs compared to standard full fine-tuning and LoRA.

**Strengths:**

- The attempt to identify reasoning-critical parameters using Fisher Information is interesting and well-motivated.

- The method is conceptually simple yet empirically effective: it improves the reasoning–safety trade-off in a consistent manner.

- The paper is clearly written and easy to follow.

- The Fisher-based parameter analysis (Sec. 4.1) provides an insightful view of where reasoning capability may reside within LLM architectures.

**Weaknesses:**

### Novelty concern – connection to prior work

The core idea of using Fisher Information to identify task-critical parameters is not new. It is fundamentally similar to [1]. However, this key prior work is not cited or discussed. The proposed method effectively reuses the same Fisher-based parameter importance estimation framework, applied here to reasoning–safety alignment. Without acknowledging this connection, the originality of the contribution appears limited.

[1] Overcoming catastrophic forgetting in neural networks,

---

### Reasoning-specific framing appears overstated
The Fisher-based importance analysis is task-agnostic—it can be applied to any loss function, not necessarily reasoning.
Thus, the “reasoning-preserved” framing feels artificially narrowed. The method could generalize to other ability (e.g., QA, translation, conversation), and the paper might be more convincing if presented as a general task-preserving fine-tuning framework rather than one specialized to reasoning and safety.

---

### Limited ablation on scaling strength
In Sec. 4.1, the authors perturb the top 1% parameters by scaling them by 0.9. It would strengthen the argument to explore multiple scaling magnitudes (e.g., {0.8, 0.9, 1.1, 1.2}), as in [2], where scaling parameters above 1.0 even amplifies the ability.

[2] SAFETY LAYERS IN ALIGNED LARGE LANGUAGE MODELS: THE KEY TO LLM SECURITY

---

### Missing comparison to [1] or related Fisher-based regularization approaches

Since [2] and similar methods also leverage mechanistic analysis to identify task-relevant parameters, a direct comparison would clarify what is unique about RPSA. If those existing methods already achieve similar or better trade-offs, the novelty of RPSA would be weakened.

---

### Generalization concerns
The Fisher estimation relies exclusively on MATH-500 as the reasoning reference dataset. It remains unclear whether similar patterns or benefits hold for other reasoning datasets (e.g., GSM8K, ARC-C, AIME, MBPP). Demonstrating cross-dataset consistency would make the claim of “reasoning-critical parameter identification” much more compelling.

### Limited backbone diversity

Experiments are only conducted on two distilled DeepSeek-R1 variants. Evaluations on larger or different model families (e.g., R1-1.5 - 32B, s1) would help assess the robustness and scalability of the approach.

**Questions:**

See weaknesses.

---

### Official Review · Reviewer_XZXa · 2025-10-31

**Soundness:** 2
**Presentation:** 3
**Contribution:** 3
**Rating:** 4
**Confidence:** 3

**Summary:**

This paper addresses the emerging safety-tax problem in Large Reasoning Models (LRMs): safety alignment often degrades reasoning performance.
The authors propose Reasoning-Preserved Safety Alignment (RPSA), which identifies and protects reasoning-critical parameters during safety fine-tuning using the diagonal Fisher Information Matrix. Parameters with high Fisher importance are selectively frozen, while the remaining ones are updated on safety data.
Experiments on DeepSeek-R1-Distill-Llama-8B and DeepSeek-R1-Distill-Qwen-7B across reasoning and safety benchmarks show that RPSA greatly reduces the safety tax—achieving comparable safety while preserving most reasoning accuracy.

**Strengths:**

1. The paper gives a sound formal analysis connecting safety degradation to interference with reasoning-critical parameters and leverages the Fisher framework as an importance estimator.  Despite technical depth, the exposition remains clear and organized, and the authors release code and append detailed prompts.

2. Freezing based on Fisher scores is conceptually clean and can be combined with both full fine-tuning and LoRA. Results demonstrate consistent improvement in the safety–reasoning trade-off.

3. Layer- and module-wise Fisher visualizations reveal that early and final layers, especially attention projections V and K, dominate reasoning importance—an observation of independent interest for future mechanistic studies

**Weaknesses:**

1. Only two mid-sized models (8B and 7B) are tested. Verification on larger or structurally different LRMs like Qwen-3 would improve generality.

2. The Fisher importance is computed on 100 MATH-500 samples; reliance on a math-centric corpus may bias.

3. Although the Fisher-based formalism is theoretically motivated, the assumptions are only heuristically justified. A deeper appendix proof or comparison with other importance metrics would clarify soundness.

4. While hyperparameters are reported, the dynamic k strategy could be elaborated.

**Questions:**

1.  Have the authors evaluated whether Fisher-based importance patterns transfer across architectures and model size? This would test whether reasoning-critical subspaces are model-family invariant.

2. Since FIM is computed on MATH-500, would using a multi-domain reasoning dataset materially change which parameters are frozen?

3. Could RPSA be integrated into reinforcement-learning-based alignment (e.g., RLVR or Safe-RLHF) as a pre-training regularizer to preserve reasoning?

4. The formulas and theoretical sections are somewhat too brief and confusing. Could the authors provide more detailed and clear definitions, formalizations, and proofs in the appendix?

I believe the authors should primarily focus on (1) adding experiments on models of more sizes and types, and (2) providing a clearer description of the motivation and theory.
If the authors address these issues and the paper's quality improves, I will consider raising my score.

---

### Official Review · Reviewer_fkws · 2025-11-03

**Soundness:** 3
**Presentation:** 3
**Contribution:** 3
**Rating:** 4
**Confidence:** 4

**Summary:**

The paper's goal is to protect the parameters that are crucial for reasoning during alignment for safety so that the resulting model is safe and show high performance in reasoning. The paper proposes to use Fisher Information Matrix to detect the importance of the parameters to reasining and freeze those during alignment on safety datasets.

**Strengths:**

- The paper touches on a very important problem, the safety tax, which degrades the current model's reasoning capabilities.

- Dynamic adjustment of "k" based on the gradient conflict between reasoning and safety objectives.

- The paper provides great insights with the Fisher Information and shows higher performance than FFT and LoRA-based alignment in Reasoning and Harmfulness metrics.

**Weaknesses:**

- The Fisher Information Matrix derived from a reasoning corpus, so it depends on how good the corpus is constructed and the generalizability method, thereby, is questionable.

- The Paper doesn't compare their approach with the other methods in the literature.

- The selection of diagonal elements is supported by the computational efficiency, yet there is no theoretical ground behind that decision.

- The proposed Fisher Information Matrix methodology simplifies to the L2 distance between the diagonal vector of parameters, which are tuned and not tuned for safety. The selection of diagonal entries is only made for computational efficiency. Therefore, this questions the novelty of the approach since many methods use L2 regularization or proximity methods for alignment.

- The reasoning capacity is measured by the selected reasoning corpus, in this case MATH-500; however, the reasoning is a generic concept, and even though the model keeps its skillful parameters for the selected reasoning corpus, it may change its parameters. This may disrupt other reasoning concepts.

**Questions:**

- Don't the authors agree that the dynamic selection of $k$ parameter during training may create instability? Because in one iteration a parameter may be considered in the gradient path, but it may not in the second iteration, which disassociates the parameters sample to sample, and the effective loss surface changes in every iteration. To me, this may cause the model to be unstable.

- How different is the obtained equation (5) from the  L2 norm between optimal $\theta_R$ and trained $\thetas$?

- What if the safety dataset is conflicting with the reasoning dataset all the time, then there won't be any updates in the parameters?

---

### Note · Authors · 2025-11-12

I have read and agree with the venue's withdrawal policy on behalf of myself and my co-authors.